# A Retrospective Study on Coinfections, Antimicrobial Resistance, and Mortality Risk Among COVID-19 Patients (2020–2021) with Consideration of Long-COVID Outcomes

**DOI:** 10.3390/microorganisms13092141

**Published:** 2025-09-12

**Authors:** Carlos Rescalvo-Casas, Rocío Fernández-Villegas, Marcos Hernando-Gozalo, Laura Seijas-Pereda, Lourdes Lledó García, Lars Arendt-Nielsen, Juan Cuadros-González, Ramón Pérez-Tanoira

**Affiliations:** 1Departamento de Biomedicina y Biotecnología, Facultad de Medicina, Universidad de Alcalá, 28805 Madrid, Spain; carlos.rescalvo@uah.es (C.R.-C.); roziofernandezville@gmail.com (R.F.-V.); spflaura@gmail.com (L.S.-P.); lourdes.lledo@uah.es (L.L.G.); juan.cuadros@uah.es (J.C.-G.); 2Departamento de Microbiología Clínica, Hospital Universitario Príncipe de Asturias, 28805 Madrid, Spain; m.hernando@uah.es; 3Departamento de Química Orgánica y Química Inorgánica, Facultad de Química, Universidad de Alcalá de Henares, 28805 Madrid, Spain; 4Center for Neuroplasticity and Pain (CNAP), Department of Health Science and Technology, School of Medicine, Aalborg University, 9000 Aalborg, Denmark; lan@hst.aau.dk; 5Department of Gastroenterology & Hepatology, Mech-Sense, Clinical Institute, Aalborg University Hospital, 9000 Aalborg, Denmark; 6Steno Diabetes Center, North Denmark, Clinical Institute, Aalborg University Hospital, 9000 Aalborg, Denmark

**Keywords:** COVID-19, coinfection, long COVID, antimicrobial susceptibility, hospitalized COVID-19 patients

## Abstract

Coinfections in COVID-19 patients can worsen disease severity by enhancing SARS-CoV-2 replication and proinflammatory cytokine levels. This study analyzes the characteristics of coinfected COVID-19 patients across the pandemic and their association with in-hospital mortality. We retrospectively examined data from 351 COVID-19 patients hospitalized in a Spanish secondary hospital between March 2020 and February–March 2021. Nasopharyngeal swabs from 340 patients were analyzed using multiplex RT-PCR to identify 26 respiratory pathogens. A total of 136 patients were coinfected with 191 bacteria (100 Gram-negative and 91 Gram-positive), 20 viruses, 18 fungi, and 1 protist. In 2021, empirical cephalosporin use increased (*p* = 0.009). The incidence of enterococcal coinfections tripled from 2020 to 2021 (*p* < 0.001). In 2021, a greater proportion of patients experienced urine (*p* = 0.001) and bloodstream (*p* = 0.010) coinfections. In 2020, there was one bloodstream infection, while in 2021, there were seven, with half of them being fatal. Coinfected patients experienced longer hospital stays and higher odds of long COVID (*p* < 0.001; *p* = 0.014; *p* = 0.045). Non-respiratory coinfections in 2021 correlated with increased mortality (*p* = 0.002). Antimicrobial resistance remained stable (*p* = 0.149). The rise in cephalosporin use correlated with increased *Enterococcus* infections, notably bloodstream infections, which were linked to mortality (*p* = 0.016). In 2021, coinfections were linked to prolonged hospital stays and an increased risk of mortality in our patient cohort.

## 1. Introduction

During the COVID-19 pandemic, coinfections with other pathogens have been observed to intensify disease severity by potentially increasing severe acute respiratory syndrome coronavirus 2 (SARS-CoV-2) replication or because a viral infection may predispose some individuals to bacterial pneumonia [1,2]. Although, early in the pandemic, bacterial coinfection rates were assumed to be high, more recent studies have shown that the actual prevalence is lower in several cohorts, highlighting the need for careful re-evaluation of empirical antibiotic use [3,4,5,6,7,8,9]. Coinfections are defined as “Any kind of pathogenic microorganism infecting the patient at the same time as the COVID test was positive”. These coinfections are associated with increased morbidity and mortality, but their prevalence, incidence, and microbiological characteristics have been poorly studied and are considered an important knowledge gap [3,4,5,6,7,8,9].

The proportion of coinfections in SARS-CoV-2-infected patients and the types of pathogens involved varied between different studies, which may be attributed to the different seasons and regions of specimen collection [2,10,11,12,13]. Hui Fan et al. highlighted that bacterial coinfections should be considered critical risk factors influencing the mobility and severity of COVID-19, with *Haemophilus influenza*, *Staphylococcus aureus*, and *Streptococcus pneumoniae* being the most common respiratory pathogens detected [14]. However, in other studies, *Klebsiella pneumoniae*, *Enterococcus faecalis* and *Acinetobacter baumannii* were the most prevalent bacterial pathogens in SARS-CoV-2 positive patients [10]. This demonstrates the variety and value of studying coinfections in COVID-19 patients, which could be a determining factor associated with in-hospital death.

Given the variability in pathogen profiles worldwide and limited local data, a better understanding of coinfections in COVID-19 patients is essential to improve clinical management and reduce inappropriate antibiotic use. In our region, data are currently not available on the microbiological spectrum of early or healthcare-associated coinfections in hospitalized COVID-19 patients. Generating such evidence is crucial to inform local clinical decision-making and to guide antimicrobial stewardship, given the potential risk of both under-treatment and inappropriate broad-spectrum antibiotic use.

Additionally, during the pandemic, there was widespread use of empirical antibiotic treatment—most commonly cephalosporins—in hospitalized COVID-19 patients [13,15,16,17]. There is a growing incidence of infections with multidrug-resistant bacterial strains during the hospitalization of patients with COVID-19, a trend frequently linked to increased mortality [17,18,19,20].

Studies reporting substantial rates of bacterial coinfections in hospitalized COVID-19 patients support the notion that such infections—and the associated inflammatory burden—may contribute to the persistence of symptoms characteristic of long COVID [11,13]. Beyond their immediate impact on disease severity, acute-phase bacterial coinfections may also contribute to post-acute sequelae. The additional inflammatory burden generated by concurrent infections can lead to prolonged immune activation, while disruption of the respiratory microbiome and episodes of immune dysregulation may impair recovery and favor the persistence of symptoms. Such mechanisms offer a plausible link between acute coinfections and long COVID, consistent with the clinical relevance of bacterial coinfections observed in hospitalized cohorts [11,13].

Thus, the present study aimed to assess the etiology of coinfections, the prevalence of multidrug-resistant bacteria, and associated patient comorbidities as potential factors related to increased mortality or the development of long COVID.

## 2. Materials and Methods

### 2.1. Study Design and Setting

This study was performed in a secondary hospital in the region of Madrid, Spain. We compared COVID-19 hospitalized cases across two different periods, March 2020 and February–March 2021, obtaining 351 patients who met the criteria for the study. To ensure comparable time periods and a consistent sample size between 2020 and 2021, only patients hospitalized without coinfection during March were included in the analysis.

The methodology employed for the selection of patients is illustrated in Figure 1. Our study focused exclusively on hospitalized adults, as pediatric and non-hospitalized populations may exhibit distinct coinfection profiles, clinical courses, and antimicrobial responses, which are not captured in the referenced studies [1,3].

This study was approved by the Ethics Committee of Hospital Universitario Príncipe de Asturias and was conducted following the updated Declaration of Helsinki; there was no need for patient consent.

### 2.2. Study Participants and Data Collection

Positive cases were determined by a positive RT-PCR result for SARS-CoV-2 in a nasopharyngeal or lower respiratory tract sample. The medical records of patients up to 12 July 2023 were reviewed for reinfection or long-COVID-19 results, following the NICE guideline criteria for long-COVID-19 determination [21].

Additionally, 95 out of the 351 individuals who died during their COVID-19 episode were not included in the analysis of reinfection and long-term COVID-19 variables. The results of microbiological studies were collected from the medical records of these 351 patients.

### 2.3. Study Variables

The following clinical data were collected from patient records:

**Hospitalization, age, sex, comorbidities** (i.e., hypertension, diabetes mellitus, coronary heart disease, chronic obstructive pulmonary disease (COPD), chronic kidney disease, and immunosuppression), **symptoms** (fever, cough, and dyspnea), **healthcare-associated infection, coinfection, reinfection, long COVID, antibiotic resistance, and empirical treatment.**

A 12-week threshold period was used to study **long COVID**, as the acute phase included symptoms presented between 4 and 12 weeks, in accordance with the NICE guidelines [21].

**Reinfection** is defined as the recurrence of COVID-19 infection after a person has previously contracted the virus, recovered from it, and then become infected again [22].

**Immunosuppression** at the time of presentation was determined by any of the following: administration of corticosteroids or another immunosuppressive drug within the previous month, ongoing antineoplastic treatment, or HIV status.

**Healthcare-associated infections** were defined as the acquisition of a disease/pathogen 48 h after being hospitalized without any evidence of a previous infection. Highly probable contamination was excluded. The following microorganisms were thus excluded:

*Corynebacterium* sp. and *Cutibacterium* sp. from blood cultures (these pathogens are usually commensal skin bacteria), *Candida* sp. from respiratory samples, and staphylococci other than *Staphylococcus aureus* and *Staphylococcus lugdunensis* from urine samples [4,5].

Pre-existing conditions, which could be symptoms misinterpreted as COVID-19, were not considered. This analysis included symptoms that were manifested concurrently with COVID-19 infection.

**Antimicrobial resistance** was associated with the number of antimicrobial drug resistance, depending on the bacteria involved and their bibliography based on phenotype.

### 2.4. Diagnostic Procedures

#### 2.4.1. Detection of COVID-19

SARS-CoV-2 infection was diagnosed according to the WHO protocol [13]. Viral RNA was obtained from clinical samples by two automatic extractors, i.e., the MagCore HF16 nucleic acid extractor (RBC Bioscience, Taipei, Taiwan) and the Hamilton Microlab Starlet liquid handler (Hamilton Company, Bonaduz, Switzerland). The extracted RNA was amplified using two real-time PCR platforms: the VI-ASURE SARS-CoV-2 Real-Time PCR Detection Kit (Certest Biotech, Saragossa, Spain) and the Allplex 2019-nCoV assay (Seegene, Seoul, Republic of Korea). These kits detect the following SARS-CoV-2 genes: N, RdRp, and E.

#### 2.4.2. Respiratory Coinfection Methods

A total of 340 respiratory samples used for the initial diagnosis of SARS-CoV-2 infection were retrieved and prospectively screened with the Allplex^TM^ Respiratory Full Panel Assay RT-PCR detection kit (Seegene, Seoul, Republic of Korea) to detect possible respiratory coinfection organisms. There were 11 samples that could not be recovered. Probable colonizations were excluded.

#### 2.4.3. Antibiotic Resistance Detection Method

Vitek2 system (BioMérieux, Nürtingen, Germany) was used for the determination of antimicrobial resistance according to EUCAST criteria [14]. From the test performed, we searched for six different antimicrobial resistance mechanisms: extended-spectrum beta-lactamase (ESBL)-producing Enterobacterales, AmpC beta-lactamases, Vancomycin-Resistant *Enterococci* (VRE), extensively drug-resistant (XDR) bacteria, OXA-48 Carbapenemase-producing Enterobacterales, and methicillin-resistant *Staphylococcus aureus*.

### 2.5. Statistical Analysis

Continuous variables are presented as medians and interquartile ranges (IQRs), while categorical variables are summarized as frequencies and percentages. To compare differences between survivors and non-survivors and coinfected versus not coinfected, we used the Mann–Whitney U-test, χ^2^ test, or Fisher’s exact test where appropriate. A statistically significant difference was considered when *p* ≤ 0.05. We performed statistical analysis using SPSS v20.0 (IBM Corp., Armonk, NY, USA). Univariate and multivariate logistic regression models were used to analyze the risk factors associated with coinfection in 2020 compared to coinfection in 2021. The logistic regression model was adjusted for the most significant variables, and the results were expressed as odds ratios (ORs) and 95% confidence intervals (CIs).

## 3. Results

### 3.1. Participant Characteristics and Comparison Between the 2020 and 2021 Cohorts

A total of 449 patients were hospitalized for COVID-19 at the Hospital Universitario Príncipe de Asturias (Madrid, Spain) during March 2020 and February–March 2021. Of these patients, 378 did not have any coinfection when admitted and met the criteria for inclusion in this study (Figure 1). The demographic and epidemiological characteristics of the included patients are presented in Table 1. In 2021, a total of 378 patients were hospitalized, of whom 109 cases were recorded in March. The cohort consisted of 57.8% males and 42.2% females, with a median age of 69.74 years (IQR: 55.03–79.64). Hospitalization rates were 26.68% in March 2020 and 29.90% in March 2021.

Over a quarter of the patients (26.76%) died during their current COVID-19 episode. Mortality was significantly higher among patients in 2020, with a mortality rate of 39.77% (n = 68), compared to 15.0% (n = 27) in 2021.

Age, ICU admission, and pneumonia prevalence were higher in the 2020 patient cohort (Table 1). On the other hand, in 2021, more patients were male (***p* = 0.032**), were immunocompromised (*p* = 0.004) with a longer hospital stay (***p* = 0.008**), had a higher prevalence of asymptomatic presentations (*p* < 0.001), and had received more empirical cephalosporin treatment (***p* = 0.009**; n = 72, 2021; n = 46, 2020) (Table 1). However, the number of empirical antibiotic treatments prescribed was similar between 2020 (n = 86) and 2021 (n = 90) (*p* = 0.956) (Table 1).

### 3.2. Analysis of Type of Coinfection 

#### 3.2.1. Distribution of Coinfecting Microorganisms Reveals Predominant Pathogens

A total of 230 pathogens were observed in 136 patients, of which 191 were bacteria (100 Gram-negative and 91 Gram-positive), 20 viruses, 18 fungi, and 1 protist. All viral infections except for one were identified in 2020 [19/20 (***p* < 0.001**)]. Table 2 shows the distribution of microorganisms according to different variables of identification.

Compared to 2020, more patients in 2021 had urinary (n = 49, 65.33%; ***p* = 0.001**) and bloodstream (n = 17, 77.27%; ***p* = 0.010**) coinfections. However, the predominance of respiratory coinfection was seen in the 2020 patient cohort (***p* = 0.022**, n = 36; 59.02%). Of the 230 pathogens detected, 21 were found to coinfect outside the respiratory tract, urine, or bloodstream (Table 2). In 2020, Gram-negative bacteria were more frequently isolated (***p* = 0.006**), whereas in 2021, more Gram-positive bacteria were detected (***p* = 0.006**).

In general, the most frequent pathogens were *Escherichia coli* [n = 34; 52.9% in 2020 vs. 47.1% in 2021], *Enterococcus faecalis* (n = 28, 28.6% in 2020 vs. 71.4% in 2021), *Enterococcus faecium* (n = 19, 21.1% in 2020 vs. 78.9% in 2021), *Staphylococcus aureus* (n = 17, 11.8% in 2020 vs. 88.2% in 2021) and *Haemophilus influenzae* (n = 15, 100% in 2020). The staphylococci other than *Staphylococcus aureus* and *Staphylococcus lugdunensis* that are presented in Table 2 were isolated from urethral catheters. The number of patients with enterococcal coinfections tripled between 2020 (n = 11) and 2021 (n = 34) (***p* < 0.001**). Only one *Enterococcus* sp. was found in blood infections in 2020 compared to seven in 2021. *Enterococcus* blood infections were associated with high mortality (***p* = 0.016**), with a 50% mortality rate.

Respiratory Full Panel Assay RT-PCR detected 38 previously undiagnosed cases of respiratory coinfection (56.72% of 67 total respiratory coinfections). The microorganisms identified with this method included 15 *Haemophilus influenzae*, 5 influenza A-H1pdm09, 6 respiratory syncytial virus A, 4 *Streptococcus pneumoniae*, 3 parainfluenza virus 4 (HPIV4), 2 rhinovirus, 2 human coronavirus OC43 (HCoV-OC43), and 1 Adenovirus (Table 2). All bacterial strains identified by the Respiratory Full Panel Assay RT-PCR were exclusively associated with patients diagnosed in the year 2020.

#### 3.2.2. Impact of Coinfections on Patient Outcomes and Prognosis

Table 3 presents the relationship between the study variables among coinfected and non-coinfected patients. In the univariate analysis, treatment with any type of antibiotic showed a protective effect against coinfection but also increased the likelihood of early reinfection (Table 3).

Additionally, coinfected patients had longer hospital stays, were immunosuppressed, and had increased odds of experiencing long COVID (Table 3). All variables found to be significant in the univariate analysis were included in the multivariate analysis. Coinfected patients were more likely to have longer hospital stays [1.077 (1.041–1.114); ***p*-value < 0.001**] than non-coinfected patients (Table 3).

### 3.3. Prevalence of Resistant Bacteria Highlights Challenges in Empirical Antibiotic Therapy

Antimicrobial resistance mechanisms (ESBL, VRE, carbapenem resistance, AMP-C, and MRSA) were studied in 131 pathogens. We found 22 microorganisms with at least one (16.79%) resistance mechanism. A higher prevalence of resistance mechanisms was observed in 2021 (15/71, 21.13% in 65 patients) compared to 2020 (7/60, 11.67% in 38 patients). However, the observed differences were not statistically significant for either type of microorganisms (*p* = 0.149) or number of individuals with bacteria showing resistance mechanisms (*p* = 0.376).

Empirical antibiotic therapy was not significantly associated with resistance mechanisms in bacterial infections (*p* = 0.749). No statistically significant difference was observed in the duration of hospital stay between patients with antibiotic-resistant infections (n = 21, median: 17.00 [10.00–34.25] days) and those with non-resistant infections (n = 82, median: 19.00 [12.50–45.50] days) (*p* = 0.328). A total of 21 cases of resistant infections were identified, with one case infected by a bacterium possessing both ESBL and OXA resistance genes. Additionally, no resistance mechanisms were found to be related to patient mortality (*p*-values: ESBL: 0.478; carbapenem resistance: 0.083; methicillin-resistant *Staphylococcus aureus* [MRSA]: 0.643; AMP-C: 0.794) (Table A1).

### 3.4. Possible Factors Associated with Mortality

Deceased patients in 2021 exhibited longer hospitalization durations, a higher number of symptoms, and higher levels of immunosuppression (***p* < 0.001**; ***p* = 0.001**; ***p* = 0.003**, respectively). Conversely, patients who died in 2020 manifested a higher incidence of pneumonia compared to survivors in 2021 (***p* = 0.012**). In 2021, 62.96% of deceased patients experienced coinfections, a rate significantly higher than the rate of 33.82% in 2020 (***p* = 0.009**). Furthermore, non-respiratory coinfections were also more frequent in deceased patients in 2021, with a statistically significant difference (***p* = 0.002**). An analysis of mortality comparing coinfected and non-coinfected patients for 2020 and 2021 shows that in 2021, patients with non-respiratory coinfections had a significantly higher likelihood of death (***p* = 0.011**).

A logistic regression analysis was conducted to explore associations between study variables and mortality (Table 4). The variables age, comorbidity, cardiovascular disease, cancer, and renal chronic disease were subjected to multivariate analysis. Among elderly patients with COVID-19, the presence of at least one comorbidity—either cardiovascular disease, cancer, or chronic renal disease—was associated with an elevated odds ratio for mortality (Table 4).

## 4. Discussion

This study aimed to investigate the impact of coinfections in SARS-CoV-2 patients and whether the etiology and antimicrobial susceptibility of pathogens changed between 2020 and 2021. Additionally, it sought to examine the effect of coinfections on patient prognosis, including mortality and long-COVID development.

### 4.1. Higher Prevalence of Coinfections in 2020 and Increased Mortality in Non-Respiratory Cases in 2021

Coinfections were more prevalent in 2020, as 65 hospitalized cases (10.14%) were identified in 641 patients in that year, compared to 71 cases (4.73%) in 1501 patients in 2021. In 2021, non-respiratory coinfections were associated with a significantly higher mortality rate when compared to patients without coinfections.

### 4.2. Shifts in Coinfection Patterns and Predominant Pathogens Between 2020 and 2021

The ratio of pathogens in coinfected patients was 1.61 per patient, which was significantly higher than that reported in the literature [1,2,3]. Russell et al. (2021) noted a high ratio of 1.08 (2109/1942) pathogens per patient [4]. The pandemic led to the use of masks and protective measures, thus reducing respiratory virus transmission. COVID-19 was the most prevalent respiratory virus during the pandemic, which may explain why there was only one single patient coinfected with another respiratory virus in 2021 [5].

While COVID-19 infections are frequently associated with other respiratory and bloodstream infections [2,3], these latter coinfections correlate with higher odds of mortality [6,7]. Notably, in 2021, blood and urinary coinfections were more common compared to 2020, possibly reflecting increased hospital-acquired infections during prolonged hospitalization. In 2021, respiratory coinfections were predominant, but blood and urinary coinfections increased, likely due to protective measures against respiratory infections [4]. In a study in Wuhan, China, involving 546 COVID-19 patients from December 2022 to January 2023, 20.18% were coinfected with bacterial pathogens, including *Haemophilus influenza*, *Staphylococcus aureus*, and *Pseudomonas aeruginosa* [14].

However, another study by Yang et al. (2024) in Shenzhen found a shift in coinfection pattern, identifying *Klebsiella pneumoniae*, *Enterococcus faecalis*, and *Acinetobacter baumannii* as the most prevalent bacterial pathogens in SARS-CoV-2-positive patients [10]. Their findings revealed that respiratory coinfections were primarily caused by bacteria and DNA viruses, while blood infections were characterized by DNA viruses [10].

Despite these insights, the full spectrum of infectious agents remains unclear, especially regarding the potential association of viruses, bacteria, and fungi with COVID-19; our study suggests that the most prevalent pathogens for respiratory coinfections are *Staphylococcus aureus* and *Haemophilus influenzae*, whereas *Staphylococcus epidermidis* and *Enterococcus faecalis* are the most prevalent for blood infections. Russel et al. (2021) and Pandey et al. (2022) reported that the prevalence of coinfection in SARS-CoV-2 and influenza varies depending on the geographical location [4,23], which could also explain the divergence in the types of coinfections in COVID-19 patients due to demographics.

The observed increase in the number of enterococcal infections can be attributed to a higher incidence of urinary tract and bloodstream infections in 2021 compared to 2020. Additionally, the empirical prescription of cephalosporins to hospitalized COVID-19-positive patients in 2021 may have contributed to the selection of *enterococci* and *E. coli* among the hospital-acquired pathogens associated with these types of infections.

Based on the multivariate analysis, we concluded that non-respiratory coinfections, which were predominant in 2021, were associated with Gram-positive bacteria, which explained the change in *enterococci* numbers. In previous studies, Gram-negative bacteria were found to be more prevalent in cases of SARS-CoV-2 with coinfections [6,11]. In fact, bloodstream coinfections were found to be predominated by Gram-positive bacteria [12]. Similarly, as observed in our study, the incidence of bloodstream coinfections significantly increased in the 2021 patient cohort [12]. Another potential explanation for this phenomenon is provided by Pandey et al. (2022), who observed an increase in empirical antibiotic coverage for the novel respiratory viral illness (COVID-19) during the pandemic [23]. This might have led to the isolation of Gram-negative bacteria later during their ICU stay, which could explain the higher prevalence of Gram-negative bacteria observed in 2020 [23].

### 4.3. Antimicrobial Resistance Patterns Remain Consistent Despite Empirical Treatment

Exposure to antimicrobials is a recognized factor that fosters bacterial resistance and is linked to increased mortality rates [24,25,26]. A total of 50% of our patient cohorts in 2020 and 2021 received empirical antibiotic treatment, but no differences in resistance mechanisms or mortality rates were observed.

In our study, we identified 16.79% of the coinfecting bacteria as exhibiting at least one resistance mechanism, which is consistent with previously reported findings [13,17]. The data suggests an increase in prevalence, possibly due to the emergence of antibiotic-resistant bacteria. To prevent further spread, antimicrobial stewardship policies may need to be relaxed, as the higher prevalence observed in 2021 may have resulted from the selection of resistant bacteria under these policies [26]. However, the apparent upward trend should be interpreted with caution due to the absence of statistical significance.

Importantly, our findings highlight differences in coinfection patterns compared to previous studies. While earlier reports often focused on specific pathogens or respiratory infections, our study demonstrates a broader spectrum of coinfecting agents, including bloodstream and urinary pathogens, and shows a shift in prevalence between 2020 and 2021. This emphasizes the dynamic nature of coinfections in hospitalized COVID-19 patients and underlines the relevance of continuous surveillance to guide empirical treatment and antimicrobial stewardship.

### 4.4. Coinfections and Comorbidities Are Associated with Increased Mortality and Long-COVID Risk

COVID-19 mortality varied considerably between the first and subsequent waves of the pandemic, suggesting a significant impact of the disease’s initial emergence on patient outcomes. Early in the pandemic, clinical characteristics such as age, ICU admission rates, and the prevalence of severe complications were more pronounced, reflecting the challenges faced in managing an unfamiliar virus.

In 2021, mortality due to SARS-CoV-2 was reduced due to the introduction of SARS-CoV-2 vaccines. However, we observed that patients remained hospitalized longer, received more empirical cephalosporin treatments, and were also more immunocompromised compared to the same period in 2020.

An analysis of deceased patients revealed similar patterns in immunosuppression, symptoms, and hospitalization. Immunosuppressed patients with COVID-19 have an increased odds ratio for death [13,27]. Additionally, a comparison between living and deceased patients showed that coinfected patients had longer hospital stays, were more immunosuppressed, and had increased odds of experiencing long COVID. This context is relevant for patients hospitalized in 2021 for other infections, with COVID-19 being detected later as part of pandemic protocols.

A final analysis was conducted to investigate the potential association between several variables and mortality rate. The results indicated that older patients and those with at least one comorbidity were more likely to die [13,28]. Upon comparing coinfected patients who had died to those who were still alive, age was the only variable that demonstrated statistically significant difference.

Furthermore, when we compared deceased patients in 2020 with those in 2021, we found that most of the deceased patients in 2021 suffered from any kind of coinfection [28]. Some studies, such as the one by Chen et al. (2022), stated that bacterial and fungal coinfections were associated with a 2.5-fold increase in the risk of death in cases with SARS-CoV-2 infection [28]. These authors hypothesized that death in these patients was caused by the collaborative involvement of SARS-CoV-2 and coinfecting pathogens in initiating a cytokine storm, potentially amplifying its impact synergistically [28]. In fact, a study conducted in Saudi Arabia showed that the pathogen *Candida albicans* had the highest CFR (Case Fatality Rate) out of all the pathogens [8].

Despite the intuitive notion that coinfection represents a significant risk factor for mortality in hospitalized patients, the findings reported by other authors accord with our results for the 2020 cohort that there is no correlation between death and coinfection in patients diagnosed with COVID-19 [11].

### 4.5. Limitations

A limitation of this study was its retrospective design, which precluded the performance of respiratory panels in all the patients. Consequently, the review of non-respiratory coinfections was conducted using patient records, which might have underestimated the actual incidence of coinfections.

The study results may be biased due to the exclusion of underage patients and non-hospitalized individuals in the patient selection process.

Finally, due to the numerous constraints inherent in this study, it would be sensible to conclude that the results obtained are applicable to the specific population under investigation. This is corroborated by previous research indicating that pathogens coinfecting with SARS-CoV-2 have a site-specific prevalence. Further studies are necessary to determine the applicability of the findings to other settings. In future studies, greater consideration will be given to the number of individuals to achieve more accurate results regarding this disease.

## 5. Conclusions

For our patient cohorts in 2020 and 2021, older hospitalized COVID-19 patients with severe symptoms, immunosuppression, and at least one comorbidity had an increased risk of death, and immunosuppressed patients were more likely to develop coinfections. The bacterial spectrum of coinfections had been reported to vary by country, year, and location, shifting from predominantly Gram-negative respiratory pathogens during the first pandemic wave to Gram-positive non-respiratory infections in 2021, likely influenced by antibiotic use. Despite these changes, the overall prevalence of antibiotic-resistant coinfecting bacteria remained stable year to year.

Further studies with different cohorts are needed to confirm that coinfections in COVID-19 patients could be a risk factor for death.

## Figures and Tables

**Figure 1 microorganisms-13-02141-f001:**
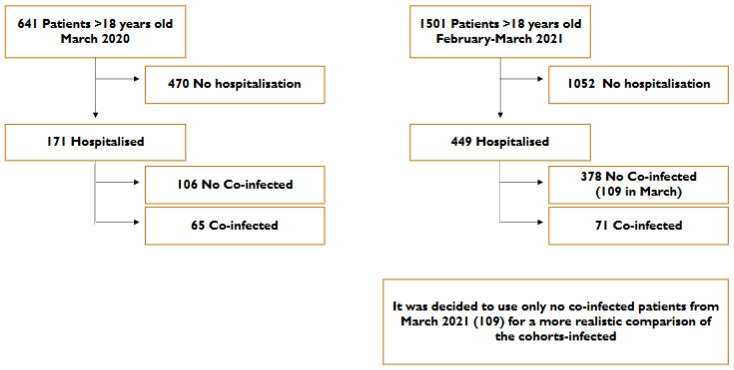
Descriptive outline of patient selection for this investigation.

**Table 1 microorganisms-13-02141-t001:** Participant characteristics (n = 351).

	2020 [n = 171 (48.7%)]	2021 [n = 180 (51.3%)]	*p*-Value
**Deceased (n = 95)**	68 (39.77%)	27 (15.00%)	**<0.001**
**Age (median)**	73.72 (61.72–82.93)	65.83 (52.40–76.79)	**<0.001**
**Sex (n = men)**	89 (52.05%)	114 (63.33%)	0.032
**Comorbidity (n = 259)**	130 (76.02%)	129 (71.67%)	0.354
**Intensive care unit (n = 23)**	20 (11.70%)	3 (1.67%)	**<0.001**
**Arterial hypertension (n = 184)**	87 (50.88%)	97 (53.89%)	0.142
**Diabetes mellitus (n = 100)**	52 (30.41%)	48 (26.67%)	0.645
**Cardiovascular disease (n = 115)**	56 (32.75%)	59 (32.78%)	0.667
**Chronic obstructive pulmonary disease (n = 58)**	31 (18.13%)	27 (15.00%)	0.574
**Cancer (n = 41)**	18 (10.53%)	23 (12.78%)	0.410
**Chronic kidney disease (n = 46)**	21 (12.28%)	25 (13.89%)	0.497
**Immunosuppression (n = 26)**	6 (3.51%)	20 (11.11%)	**0.004**
**Long-COVID (n = 62)**	30 (17.54%)	32 (17.78%)	0.157
**Reinfection (n = 37)**	11 (6.43%)	26 (14.44%)	0.163
**COVID symptoms (n = 327)**	170 (99.42%)	157 (87.22%)	**<0.001**
**Days hospitalized**	9 (5.00–16.00)	11 (7.00–23.75)	**0.008**
**Pneumonia (n = 246)**	143 (83.63%)	106 (58.89%)	**<0.001**
**Fever (n = 109)**	122 (71.35%)	87 (48.33%)	**0.002**
**Cough (n = 173)**	106 (62.00%)	67 (37.22%)	**<0.001**
**Empirical antibiotic (n = 176)**	86 (50.29%)	90 (50.00%)	0.956
**Empirical cephalosporin treatment (n = 118)**	46 (26.90%)	72 (40.00%)	**0.009**

**Table 2 microorganisms-13-02141-t002:** Description of microorganisms involved in coinfection. (**A**) Pathogens infecting patients in 2020. (**B**) Pathogens infecting patients in 2021. * Others including infections in exudates, peritoneal fluids, feces, abscesses, and joint tissues.

(A)													
	Pathogen	Onset Type				Site of Infection							
	2020	Community Onset	Row %	Hospital-Onset	Row %	Respiratory	Row %	Non-Respiratory					
								Blood	Row%	Urine	Row %	Other *	Row %
**VIRUS**	**Coronavirus OC43**	0	0	2	50	2	50	0	0	0	0	0	0
	**Rhinovirus**	0	0	2	50	2	50	0	0	0	0	0	0
	**Flu A-H1pdm09**	0	0	5	50	5	50	0	0	0	0	0	0
	**Parainfluenza virus 4**	0	0	3	50	3	50	0	0	0	0	0	0
	**Respiratory syncytial virus A**	1	8.33	5	41.67	6	50	0	0	0	0	0	0
**FUNGUS**	** *Candida albicans* **	3	42.86	0	0	0	0	1	14.29	2	28.57	1	14.29
	** *Candida parapsilosis* **	1	50	0	0	0	0	0	0	1	50	0	0
	** *Candida glabrata* **	0	0	0	0	0	0	0	0	0	0	1	100
	** *Aspergillus fumigatus* **	1	50	0	0	1	50	0	0	0	0	0	0
**GRAM − Bacteria**	** *Escherichia coli* **	6	18.75	8	25	1	3.13	1	3.13	12	37.5	4	12.5
	** *Pseudomonas aeruginosa* **	2	28.57	1	14.29	1	14.29	1	14.29	1	14.29	1	14.29
	** *Klebsiella pneumoniae* **	3	30	2	20	1	10	1	10	3	30	0	0
	** *Proteus hauseri* **	0	0	1	50	0	0	0	0	1	50	0	0
	** *Providenza stuarti* **	0	0	1	50	0	0	0	0	1	50	0	0
	** *Serratia marcescens* **	1	16.67	2	33.33	2	33.33	0	0	1	16.67	0	0
	** *Campylobacter jejuni* **	0	0	1	50	0	0	1	50	0	0	0	0
	** *Stenotrophomonas maltofilia* **	1	50	0	0	1	50	0	0	0	0	0	0
	** *Serratia species* **	1	50	0	0	1	50	0	0	0	0	0	0
	** *Haemofilus influenza* **	0	0	15	48.39	15	48.39	0	0	0	0	1	3.23
	** *Bacillus cereus* **	0	0	0	0	0	0	0	0	0	0	1	100
	** *Klebsiella oxytoca* **	0	0	0	0	0	0	0	0	0	0	0	0
	** *Proteus mirabilis* **	0	0	0	0	0	0	0	0	0	0	1	100
**GRAM + Bacteria**	** *Enterococcus faecalis* **	9	29.03	6	19.35	0	0	1	3.23	14	45.16	1	3.23
	** *Enterococcus faecium* **	11	23.91	12	26.09	0	0	0	0	23	50	0	0
	** *Staphylococcus epidermidis* **	2	50	0	0	1	25	0	0	1	25	0	0
	** *Streptococcus agalactie* **	0	0	1	50	0	0	0	0	1	50	0	0
	** *Aerococcus viridans* **	0	0	1	50	0	0	0	0	1	50	0	0
	** *Staphylococcus hominis* **	1	50	0	0	0	0	0	0	1	50	0	0
	** *Listeria monocytogenes* **	1	50	0	0	0	0	1	50	0	0	0	0
	** *Streptococcus pnuemoniae* **	0	0	5	50	4	40	1	10	0	0	0	0
	** *Staphylococcus aureus* **	0	0	0	0	0	0	0	0	0	0	3	100
	** *Clostridioides difficile* **	0	0	0	0	0	0	0	0	0	0	2	100
**(B)**													
	**Pathogen**	**Onset Type**				**Site of Infection**							
	**2021**	**Community Onset**	**Row** **%**	**Hospital-Onset**	**Row** **%**	**Respiratory**	**Row** **%**	**Non-Respiratory**					
								**Blood**	**Row** **%**	**Urine**	**Row** **%**	**Other**	**Row** **%**
**VIRUS**	** *Adenovirus* **	0	0	1	50	1	50	0	0	0	0	0	0
**FUNGUS**	** *Candida albicans* **	4	40	1	10	0	0	0	0	5	50	0	0
	** *Candida glabrata* **	1	50	0	0	0	0	0	0	1	50	0	0
	** *Aspergillus fumigatus* **	5	50	0	0	5	50	0	0	0	0	0	0
	***Aspergillus* sp.**	1	50	0	0	1	50	0	0	0	0	0	0
**Parasites**	** *Blastocistis hominis* **	0	0	0	0	0	0	0	0	0	0	1	100
**GRAM − Bacteria**	** *Escherichia coli* **	9	26.47	7	20.59	3	8.82	3	8.82	10	29.41	2	5.88
	** *Enterobacter aerobius* **	1	50	0	0	0	0	0	0	1	50	0	0
	** *Enterobacter aerogenes* **	2	50	0	0	0	0	0	0	2	50	0	0
	** *Enterobacter cloacae* **	1	33.33	0	0	0	0	1	33.33	0	0	1	33.33
	** *Pseudomonas aeruginosa* **	4	20	5	25	4	20	1	5	4	20	2	10
	** *Klebsiella pneumoniae* **	1	14.29	2	28.57	1	14.29	1	14.29	1	14.29	1	14.29
	** *Cibrobacter kosseri* **	1	50	0	0	0	0	0	0	1	50	0	0
	** *Proteus mirabilis* **	0	0	1	50	0	0	1	50	0	0	0	0
	** *Stenotrophomonas maltofilia* **	1	25	1	25	2	50	0	0	0	0	0	0
	** *Serratia marcescens* **	1	50	0	0	1	50	0	0	0	0	0	0
	***Klebsiella* sp.**	1	16.67	2	33.33	3	50	0	0	0	0	0	0
	** *Moraxella catarrhalis* **	1	50	0	0	1	50	0	0	0	0	0	0
	** *Mycobacterium lentiflavus* **	0	0	1	50	1	50	0	0	0	0	0	0
	** *Achromobacter dentrificans* **	1	50	0	0	1	50	0	0	0	0	0	0
	** *Achromobacter xyloxidans* **	1	50	0	0	1	50	0	0	0	0	0	0
	** *Raoutella ornhithinoli* **	1	50	0	0	1	50	0	0	0	0	0	0
	** *Bacteroides fragilis* **	0	0	0	0	0	0	0	0	0	0	1	100
**GRAM + Bacteria**	** *Enterococcus faecalis* **	12	31.57	7	18.42	0	0	5	13.16	14	36.84	0	0
	** *Enterococcus faecium* **	13	41.94	2	6.45	0	0	2	6.45	13	41.95	1	3.22
	** *Staphylococcus epidermidis* **	6	37.5	2	12.5	0	0	7	43.75	1	6.25	0	0
	** *Streptococcus agalactie* **	1	50	0	0	0	0	0	0	1	50	0	0
	** *Staphylococcus haemolyticus* **	0	0	1	50	0	0	0	0	1	50	0	0
	** *Staphylococcus aureus* **	10	32.26	4	12.90	11	35.48	3	9.68	0	0	3	9.68
	** *Staphylococcus hominis* **	1	50	0	0	0	0	1	50	0	0	0	0
	** *Streptococcus mitis* **	1	50	0	0	0	0	1	50	0	0	0	0
	** *Clostridioides difficile* **	0	0	0	0	0	0	0	0	0	0	1	100

**Table 3 microorganisms-13-02141-t003:** Comparative analysis of coinfected patients versus non-coinfected patients.

	Non-Coinfected Patients (n = 215)	Coinfected Patients (n = 136)	*p*-Value (χ^2^)	Univariate	*p*-Value	Multivariate (Odds Ratio, IC:95%)	*p*-Value
**Age**	68.32 (54.56–78.88)	72.11 (57.13–81.28)	0.117				
**Days hospitalized**	8 (4.00–13.00)	16 (9.00–33.00)	**<0.001**	**1.076** **(1.053–1.110)**	**<0.001**	**1.077** **(1.041–1.114)**	**<0.001**
**Empirical antibiotic (n = 176)**	121 (56.28%)	55 (40.44%)	**0.004**	**0.527** **(0.341–0.816)**	**0.004**	1.077 (0.121–1.127)	0.080
**Empirical cephalosporin (n = 55)**	84 (39.07%)	34 (25.00%)	**0.007**	**0.520** **(0.323–0.836)**	**0.007**	0.665 (0.217–2.042)	0.476
**Deceased (n = 95)**	55 (25.58%)	40 (29.41%)	0.431				
**Sex (men, n = 203)**	126 (58.60%)	77 (56.62%)	0.713				
**Comorbidity** **(n = 259)**	148 (68.84%)	111 (81.62%)	**0.008**	**2.010** **(1.194–3.385)**	**0.009**	*	
**Arterial hypertension (n = 184)**	103 (47.01%)	81 (59.56%)	0.553				
**Diabetes mellitus (n = 100)**	51 (23.72%)	49 (36.03%)	0.113				
**Cardiovascular disease (n = 115)**	62 (28.84%)	53 (38.97%)	0.348				
**Chronic obstructive pulmonary disease (n = 58)**	32 (14.88%)	26 (19.12%)	0.731				
**Cancer (n = 41)**	21 (9.77%)	20 (14.71%)	0.403				
**Chronic kidney disease (n = 46)**	24 (11.16%)	22 (16.18%)	0.453				
**Immunosuppression** **(n = 26)**	9 (4.19%)	17 (12.50%)	**0.014**	**2.793** **(1.195–6.530)**	**0.018**	1.816 (0.447–7.373)	0.404
**Symptoms** **(n = 327)**	200 (93.02%)	127 (93.38%)	0.897				
**Pneumonia** **(n = 269)**	165 (76.74%)	84 (61.76%)	**0.001**	**0.414** **(0.247–0.695)**	**0.001**	0.487 (0.200–1.184)	0.113
**Fever** **(n = 209)**	136 (63.26%)	73 (53.68%)	0.054				
**Cough** **(n = 173)**	115 (53.48%)	58 (42.65%)	**0.037**	**0.621** **(0.397–0.972)**	**0.037**	0.709 (0.324–1.551)	0.389
**Long COVID** **(n = 62)**	32 (14.88%)	30 (22.06%)	**0.045**	**1.804** **(1.010–3.222)**	**0.046**	1.773 (0.710–4.432)	0.220
**ICU (Intensive Care Unit) (n = 97)**	17 (7.91%)	6 (4.41%)	0.197				
**Reinfection (n = 37)**	30 (13.95%)	7 (5.15%)	**0.008**	**0.322** **(0.136–0.767)**	**0.010**	0.364 (0.087–1.528)	0.168

* The chi-square statistic is the difference in the −2 log likelihoods between the final model and the reduced model. The reduced model is created by omitting an effect from the final model. The null hypothesis is that all the parameters of that effect are 0. This reduced model is equivalent to the final model since the omission of the effect does not increase the degree of freedom; specifically, the omission or inclusion of the variable “Comorbidity” does not have any effect on the results of the other variables.

**Table 4 microorganisms-13-02141-t004:** Comparison of living patients against all deceased patients.

	Alive Patients (n = 256)	Deceased Patients (n = 95)	*p*-Value (χ^2^)	Univariate	*p*-Value	Multivariate (Odds Ratio, IC:95%)	*p*-Value
**Age**	65.29 (51.23–74.49)	80.91 (74.31–87.52)	**<0.001**	**1.077** **(1.054–1.100)**	**<0.001**	**1.049** **(1.025–1.075)**	**<0.001**
**Days hospitalized**	11 (6–19.75)	9.00 (6–16)	0.052	0.981 (0.963–1.000)	**0.047**	**0.954** **(0.927–0.980)**	**0.001**
**Empirical antibiotic (n = 176)**	128 (50.00%)	48 (50.53%)	0.930	1.021 (0.638–1.636)	0.930		
**Empirical cephalosporin (n = 55)**	29 (11.33%)	26 (27.37%)	0.131	0.672 (0.400–1.128)	0.132		
**Sex (men, n = 203)**	149 (58.20%)	54 (56.84%)	0.819	1.057 (0.657–1.702)	0.819		
**Comorbidity** **(n = 259)**	174 (67.97%)	85 (89.47%)	**<0.001**	4.006 (1.978–8.114)	**<0.001**	*	
**Arterial hypertension** **(n = 184)**	121 (47.27%)	63 (66.32%)	0.446	1.254 (0.700–2.247)	0.446		
**Diabetes mellitus** **(n = 100)**	63 (24.61%)	37 (38.95%)	0.256	1.358 (0.801–2.304)	0.256		
**Cardiovascular disease** **(n = 115)**	64 (25.00%)	51 (53.68%)	**<0.001**	**2.578** **(1.514–4.390)**	**<0.001**	**2.153** **(1.169–3.962)**	**0.014**
**Chronic obstructive** **pulmonary disease (n = 58)**	35 (13.67%)	23 (24.21%)	0.208	1.473 (0.804–2.699)	0.210		
**Cancer (n = 41)**	22 (8.59%)	19 (20.00%)	**0.044**	1.989 (1.009–3.920)	**0.047**	**2.741** **(1.256–5.980)**	**0.011**
**Chronic kidney disease (n = 46)**	25 (9.77%)	21 (22.11%)	**0.041**	1.956 (1.021–3.746)	**0.043**	1.736 (0.812–3.713)	0.155
**Immunosuppression (n = 26)**	16 (6.25%)	10 (10.53%)	0.518	1.317 (0.570–3.039)	0.519		
**Symptoms (n = 327)**	236 (92.19%)	91 (95.79%)	0.235	1.928 (0.641–5.794)	0.242		
**Pneumonia (n = 269)**	174 (67.97%)	95 (100.00%)	0.098	1.670 (0.905–3.082)	0.101		
**Fever (n = 209)**	153 (59.77%)	56 (58.95%)	0.579	0.868 (0.527–1.431)	0.579		
**Cough (n = 173)**	123 (48.05%)	50 (52.63%)	0.646	1.120 (0.689–1.821)	0.646		
**Respiratory** **coinfection (n = 67)**	49 (19.14%)	18 (18.95%)	0.967	0.988 (0.542–1.800)	0.967		
**No respiratory** **coinfection (n = 97)**	67 (26.17%)	30 (31.58%)	0.324	1.295 (0.774–2.167)	0.325		
**Blood infection (n = 22)**	22 (8.59%)	7 (7.37%)	0.388	1.385 (0.660–2.905)	0.389		
**Both coinfections (n = 28)**	20 (7.81%)	8 (8.42%)	0.859	1.080 (0.459–2.543)	0.859		
**Antimicrobial resistance (n = 103)**	15 (5.86%)	6 (6.32%)	0.864	0.912 (0.317–2.625)	0.864		
**ESBL (n = 56)**	6 (2.34%)	6 (6.32%)	0.478	1.005 (0.992–1.018)	0.480		
**Vancomycin** **-** **resistant** ***enterococcus* (n = 24)**	0 ( 0.00%)	0 (0.00%)	.	-			
**Carbapenem resistance** **(n = 56)**	4 (1.56%)	0 (0.00%)	0.084	0.811 (0.000–19.370)	0.999		
**AMP-C (n = 56)**	1 (0.39%)	1 (1.05%)	0.794	1.004 (0.976–1.033)	0.795		
**MRSA (n = 17)**	5 (1.95%)	1 (1.05%)	0.643	1.007 (0.977–1.037)	0.648		

* The chi-square statistic is the difference in the −2 log likelihoods between the final and reduced models. The reduced model is created by omitting an effect from the final model. The null hypothesis is that all the parameters of that effect are 0. This reduced model is equivalent to the final model since the omission of the effect does not increase the degree of freedom; specifically, the omission or inclusion of the variable “Comorbidity” does not have any effect on the results of the other variables.

## Data Availability

The data presented in this study are available from the corresponding author upon request, as the data cannot be shared publicly because they are confidential. The data are available from the Departamento de Microbiología Clínica, Hospital Universitario Príncipe de Asturias, for researchers who meet the criteria to access confidential data.

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
