# Peer review of "A Retrospective Study on Coinfections, Antimicrobial Resistance, and Mortality Risk Among COVID-19 Patients (2020–2021) with Consideration of Long-COVID Outcomes"

_microorganisms, 2025, doi:10.3390/microorganisms13092141_

Round 1
Reviewer 1 Report
Comments and Suggestions for Authors
- The introduction could briefly note the discrepancy between early pandemic assumptions (high bacterial co-infection rates) and more recent findings (lower than initially estimated in some cohorts).
- The current introduction mentions long-COVID in the title but only minimally in the text. A reviewer might flag this as inconsistent.
- add paragraph summarizing possible mechanisms linking acute-phase co-infections to post-acute sequelae (e.g., prolonged inflammation, microbiome disruption, immune dysregulation).
- The “no data is currently available” statement should explicitly mention your region/country and explain why the findings are important for local clinical decision-making and antimicrobial stewardship.
- Fix typographical errors ("se ing" to "setting", "HF16nucleic" to "HF16 nucleic").
- Clearly distinguish between probable colonization, contamination, and true infection.
- Specify whether antimicrobial resistance patterns were based on phenotypic testing only, or if genotypic methods were used.
- Include the exact gene targets for SARS-CoV-2 PCR (e.g., N, RdRp, E).
- Indicate whether percentages are row percentages (within pathogen) or column percentages (within year).
- The prevalence of resistant isolates (16.79%) is consistent with other studies, but the trend towards increase should be interpreted carefully given the lack of statistical significance.
- The exclusion of pediatric and non-hospitalized populations is important, but you could elaborate on how the co-infection profile might differ in these groups, based on existing literature. This would help readers understand the extent to which your findings are limited to hospitalized adults.
- Emphasize what is unique about your findings compared with prior literature, especially regarding temporal pathogen shifts and resistance stability across years.
- Consider clarifying the selection criteria for included patients.
Author Response
- the introduction could briefly note the discrepancy between early pandemic assumptions (high bacterial co-infection rates) and more recent findings (lower than initially estimated in some cohorts).
The following text has been added from the line 45 until 48. Although early in the pandemic bacterial co-infection rates were assumed to be high, more recent studies have shown that the actual prevalence is lower in several cohorts, highlighting the need for careful re-evaluation of empirical antibiotic use [3–9]
- The current introduction mentions long-COVID in the title but only minimally in the text. A reviewer might flag this as inconsistent.
A new phrase has been added to the introduction to avoid this problem : “Studies reporting substantial rates of bacterial coinfection in hospitalized COVID-19 patients support the notion that such infections—and the associated inflammatory bur-den—may contribute to the persistence of symptoms characteristic of long COVID [13,11]. Line 73-75.
- add paragraph summarizing possible mechanisms linking acute-phase co-infections to post-acute sequelae (e.g., prolonged inflammation, microbiome disruption, immune dysregulation).
Beyond their immediate impact on disease severity, acute-phase bacterial co-infections may also contribute to post-acute sequelae. The additional inflammatory burden generated by concurrent infections can lead to prolonged immune activation, while disruption of the respiratory microbiome and episodes of immune dysregulation may impair recovery and favor the persistence of symptoms. Such mechanisms offer a plausible link between acute co-infections and long COVID, consistent with the clinical relevance of bacterial coinfections observed in hospitalized cohorts [13,11]. Line 76-82
- The “no data is currently available” statement should explicitly mention your region/country and explain why the findings are important for local clinical decision-making and antimicrobial stewardship.
The phase has been reformulated with this:
In our region, no data are currently available on the microbiological spectrum of early or healthcare-associated co-infections in hospitalized COVID-19 patients. Generating such evidence is crucial to inform local clinical decision-making and to guide antimicrobial stewardship, given the potential risk of both under-treatment and inappropriate broad-spectrum antibiotic use.
- Fix typographical errors ("se ing" to "setting", "HF16nucleic" to "HF16 nucleic").
Fixed in the text
- Clearly distinguish between probable colonization, contamination, and true infection.
It is explained in the text in the following phrases “Healthcare associated infections were defined as the acquisition of a dis-ease/pathogen 48 hours after being hospitalised without any evidence of previous infec-tion. Highly probable contamination was excluded. The microorganisms thus excluded were:
Corynebacterium sp and Cutibacterium sp from blood cultures (they are usually com-mensal skin bacteria), Candida sp from respiratory samples and staphylococci other than Staphylococcus aureus and Staphylococcus lugdunensis from urine samples [4,5].”
- Specify whether antimicrobial resistance patterns were based on phenotypic testing only, or if genotypic methods were used.
Only phenotypic methods. “All this antimicrobial resistance were associated to the number of antimicrobial drugs resistance depending on the bacteria and bibliography based on phenotype” Lines 133-134
- Include the exact gene targets for SARS-CoV-2 PCR (e.g., N, RdRp, E).
Included in the line 143.
- Indicate whether percentages are row percentages (within pathogen) or column percentages (within year).
The table has been modified.
- The prevalence of resistant isolates (16.79%) is consistent with other studies, but the trend towards increase should be interpreted carefully given the lack of statistical significance.
To interpret it carefully the following sentence has been added: “Although the apparent upward trend should be interpreted with caution due to the absence of statistical significance.”
- The exclusion of paediatric and non-hospitalized populations is important, but you could elaborate on how the co-infection profile might differ in these groups, based on existing literature. This would help readers understand the extent to which your findings are limited to hospitalized adults.
To clarify this regard, the following sentence has been added Our study focused exclusively on hospitalized adults, as pediatric and non-hospitalized populations may exhibit distinct co-infection profiles, clinical courses, and antimicrobial responses, which are not captured in the referenced studies [1,3].
- Emphasize what is unique about your findings compared with prior literature, especially regarding temporal pathogen shifts and resistance stability across years.
The text has been added to the last part of the 4.3 point t: Importantly, our findings highlight differences in co-infection patterns compared to previous studies. While earlier reports often focused on specific pathogens or respiratory infections, our study demonstrates a broader spectrum of co-infecting agents, including bloodstream and urinary pathogens, and shows a shift in prevalence between 2020 and 2021. This emphasizes the dynamic nature of co-infections in hospitalized COVID-19 patients and underlines the relevance of continuous surveillance to guide empirical treatment and antimicrobial stewardship.
- Consider clarifying the selection criteria for included patients.
The selection criteria has been re-writed to make it more clear :
The study was conducted in a secondary hospital in the Madrid region, Spain. We compared hospitalized COVID-19 cases across two periods: March 2020 and February–March 2021, including a total of 351 patients who met the study criteria. To ensure comparable time frames and consistent sample sizes between 2020 and 2021, only patients hospitalized in March without co-infections were included in the analysis.
Reviewer 2 Report
Comments and Suggestions for Authors
This manuscript by Rescalvo-Casas et al. examines co-infections in two cohorts of COVID-19 patients: March 2020 and March 2021. They investigate associations with immunosuppression, comorbidity, fatality and duration of symptoms. The antibiotic resistance of the identified bacteria is also assessed. They find similar rates of hospitalisation between cohorts but a higher mortality in 2020 compared to 2021, likely due to vaccine availability. The 2021 cohort also has a much lower incidence of respiratory virus co-infection, possibly due to the uptake of masking and other protective measures during the pandemic. They see more respiratory, gram-negative bacterial co-infection in 2020 and more bloodstream, gram-positive in 2021. Combining the cohorts, Rescalvo-Casas et al. look at age, incidence of co-morbidities and fatality in co-infected vs. non co-infected patients. The conclusions lack clarity and sections of this paper that need to be rephrased but overall it contains interesting information that adds to the pool of COVID-19 cohort data.
Methods
Lines 118-119 need to be rephrased, I don’t know what you are trying to say.
Line 150 "with in-co-infection..." should read "with co-infection"
Results
Lines 155-157 to be removed
Line 162 it looks like you had 449 hospitalised with COVID-19 and 378 of them had no co-infection. You should clarify this here.
Lines 191-192 state the comparison " ... more patients suffering ... in 2021 patients" e.g. by adding "compared to 2020 patients"
Line 205 remove "with four of them dying (50%)" or re-phrase as "with a 50 % mortality rate”.
Section headings should be re-written to reflect the conclusions rather than being general "comparison between X and Y"
Section 3.4 instead of referring to "dead patients" rework this section to refer to associations with mortality/fatality. or alive vs. deceased.
Lines 273-274 rewrite to state the % of patients hospitalised and co-infected for 2020 vs. 2021.
Lines 248-254 which table compares alive and. deceased in 2021 vs. 2022? Table 4 appears to contain all patients grouped together so I can't work out where your results for this paragraph are coming from. The same applies for lines 263-265.
Discussion
Lines 277-280 are too strongly worded. I do not believe that this paper shows that respiratory pathogens were underdiagnosed due to healthcare collapse because of the pandemic.
Lines 289-291 do not make sense - in 2021 respiratory co-infections dominated due to protective measures against respiratory viruses? Blood and urinary co-infectious were increased compared to what?
The dot points in your conclusions section should be the titles of the results sections. Remove the conclusions section or write a short paragraph summarising everything.
Lines 395-396 Variation by country or geographic location is more of a discussion point than a conclusion to this study.
Table S1 appears to be the only source of results described in section 3.3 and so should be part of the main body of the manuscript.
Lines 435-436 none of the acronyms listed in this table appear anywhere in the body of the manuscript. Remove.
Comments on the Quality of English Language
The phrasing of this manuscript requires improvement before it can communicate your findings clearly enough for publication.
Author Response
This manuscript by Rescalvo-Casas et al. examines co-infections in two cohorts of COVID-19 patients: March 2020 and March 2021. They investigate associations with immunosuppression, comorbidity, fatality and duration of symptoms. The antibiotic resistance of the identified bacteria is also assessed. They find similar rates of hospitalisation between cohorts but a higher mortality in 2020 compared to 2021, likely due to vaccine availability. The 2021 cohort also has a much lower incidence of respiratory virus co-infection, possibly due to the uptake of masking and other protective measures during the pandemic. They see more respiratory, gram-negative bacterial co-infection in 2020 and more bloodstream, gram-positive in 2021. Combining the cohorts, Rescalvo-Casas et al. look at age, incidence of co-morbidities and fatality in co-infected vs. non co-infected patients. The conclusions lack clarity and sections of this paper that need to be rephrased but overall, it contains interesting information that adds to the pool of COVID-19 cohort data.
Methods
Lines 118-119 need to be rephrased, I don’t know what you are trying to say.
It has been clarified.
Line 150 "with in-co-infection..." should read "with co-infection"
The problem has been resolved.
Results
Lines 155-157 to be removed
Those lines have been removed.
Line 162 it looks like you had 449 hospitalised with COVID-19 and 378 of them had no co-infection. You should clarify this here.
The paragraph where the line162 is has been adequately changed to clarify this
A total of 449 patients were hospitalized for COVID-19 at the Hospital Universitario Príncipe de Asturias (Madrid, Spain) during March 2020 and February–March 2021. Of these, 378 patients did not have any co-infection and met the criteria for inclusion in this study (Figure 1). Demographic and epidemiological characteristics of the included patients are presented in Table 1. In 2021, a total of 378 patients were hospitalized, of whom 109 cases were recorded in March. The cohort consisted of 57.8% male and 42.2% female, with a median age of 69.74 years (IQR: 55.03–79.64). Hospitalization rates were 26.68% in March 2020 and 29.90% in March 2021.
Lines 191-192 state the comparison " ... more patients suffering ... in 2021 patients" e.g. by adding "compared to 2020 patients"
It has been changed to:
“Compared to 2020, more patients in 2021 had urinary (n = 49, 65.33%; p=0.001) or bloodstream (n = 17, 77.27%; p=0.010) co-infections”
Line 205 remove "with four of them dying (50%)" or re-phrase as "with a 50 % mortality rate”.
Change has been made.
Section headings should be re-written to reflect the conclusions rather than being general "comparison between X and Y"
The section headings that are general has been adequately changed.
Section 3.4 instead of referring to "dead patients" rework this section to refer to associations with mortality/fatality. or alive vs. deceased.
The sections has been re-written to refer to those associations.
Lines 273-274 rewrite to state the % of patients hospitalised and co-infected for 2020 vs. 2021.
Percentage of patients hospitalised has been added. “Co-infections were more prevalent in 2020, as 65 hospitalized cases (10.14%) were identified among 641 patients, compared to 71 cases (4.73%) among 1,501 patients in 2021. In 2021, non-respiratory co-infections were associated with a significantly higher mortality rate compared to patients without co-infections”
Lines 248-254 which table compares alive and. deceased in 2021 vs. 2022?
A table comparing these data was initially included; however, after further consideration, it was removed because the information could be presented more clearly and effectively within the text.
Table 4 appears to contain all patients grouped together so I can't work out where your results for this paragraph are coming from. The same applies for lines 263-265.
A table displaying these data was initially included but was later removed, as the information could be communicated more clearly and effectively within the text. Additionally, the section has been reorganized to enhance clarity and improve the flow of results.
Discussion
Lines 277-280 are too strongly worded. I do not believe that this paper shows that respiratory pathogens were underdiagnosed due to healthcare collapse because of the pandemic.
The paragraph referring to that part has been removed.
Lines 289-291 do not make sense - in 2021 respiratory co-infections dominated due to protective measures against respiratory viruses? Blood and urinary co-infectious were increased compared to what?
We thank the reviewer for pointing this out. The revised text now reads: While COVID-19 infections are frequently associated with other respiratory and bloodstream infections [2,3], these co-infections correlate with higher mortality odds [6,7]. Notably, in 2021, blood and urinary co-infections were more common compared to 2020, possibly reflecting increased hospital-acquired infections during prolonged hospitalizations
The dot points in your conclusions section should be the titles of the results sections. Remove the conclusions section or write a short paragraph summarising everything.
The titles of the results sections have been changed to a more suitable title describing the results obtained. The conclusion has been summarised like:
Older hospitalized COVID-19 patients with symptoms, immunosuppression, and at least one comorbidity have an increased risk of death, and immunosuppressed patients are more likely to develop co-infections. The bacterial spectrum of co-infections varied by country, year, and location, shifting from predominantly gram-negative respiratory path-ogens during the first pandemic wave to gram-positive non-respiratory infections in 2021, likely influenced by antibiotic use. Despite these changes, the overall prevalence of antibi-otic-resistant co-infecting bacteria remained stable year-to-year.
Lines 395-396 Variation by country or geographic location is more of a discussion point than a conclusion to this study.
That phrase was removed from the main test.
Table S1 appears to be the only source of results described in section 3.3 and so should be part of the main body of the manuscript.
Table S1 is included as supplementary material solely to provide a clearer visual representation of the data. All important results it contains are fully described in Section 3.3 of the manuscript.
Lines 435-436 none of the acronyms listed in this table appear anywhere in the body of the manuscript. Remove.
They were removed.